# Microscopic Pyrolytic and Electric Decomposition Mechanism of Insulating Polyimide/Boron Nitride Nanosheet Composites based on ReaxFF

**DOI:** 10.3390/polym14061169

**Published:** 2022-03-15

**Authors:** Xiaosong Wang, Tong Zhao, Yihan Wang, Li Zhang, Liang Zou

**Affiliations:** 1School of Electrical Engineering, Shandong University, Jinan 250061, China; wangxiaosong@mail.sdu.edu.cn (X.W.); zhleee@sdu.edu.cn (L.Z.); zouliang@sdu.edu.cn (L.Z.); 2State Grid Jinan Power Supply Company, Jinan 250010, China; wangyihan@mail.sdu.edu.cn

**Keywords:** polyimide, boron nitride nanosheets, molecular dynamics, pyrolytic, electric decomposition, reactive species in plasma

## Abstract

High thermal conductivity insulating materials with excellent comprehensive properties can be obtained by doping boron nitride nanosheets (BNNSs) into polyimide (PI). To study the microscopic mechanism of composite material decomposition in an actual working environment and the inhibitory effect of BNNS doping on the decomposition process, molecular dynamics simulations were carried out at high temperatures, in intense electric fields, and with various reactive species in plasma based on the reactive force field (ReaxFF). The results showed that the decomposition was mainly caused by hydrogen capture and adsorption, which broke the benzene ring and C-N bond on the PI chains and led to serious damage to the PI structure. The BNNS filling was shown to inhibit the decomposition of the PI matrix at high temperatures and in intense electric fields. Moreover, the BNNS filling also inhibited the material decomposition caused by ·OH and ·NO. The erosive effect of the positive corona on the PI composites was more obvious than that of the negative corona. In this paper, the microscopic dynamic reaction paths of material pyrolysis in various environments were revealed at the atomic level, and it was concluded that BNNS doping could effectively inhibit the decomposition of PI in various environments.

## 1. Introduction

As a new type of polymer insulation material, polyimide (PI) has excellent characteristics, such as its high insulation strength, low dielectric constant, corrosion resistance, and low thermal expansion [1,2]. Therefore, PI is widely used in high-voltage electrical equipment (e.g., high-frequency transformers, variable-frequency traction motors, and solid-state transformers), the aerospace industry (e.g., fairings, missile shells, and space vehicles), microelectronics fields (e.g., substrates and packages for electronic devices), and other high-tech fields [3,4,5,6]. In actual operation, PI materials face ultrahigh temperatures and the accumulation of immense heat due to various factors, including the large current in high-voltage equipment and the fast and dense circuit structure in microelectronic devices. Therefore, in addition to excellent insulating performance, a PI material is also required to possess good thermal stability and thermal conductivity. However, pure PI material tends to crack and age at high temperatures due to its poor thermal stability. Additionally, the structural regularity of the PI macromolecule is very low. The nonharmonic vibration of PI molecules and its lattice-like structure result in plenty of phonon scattering inside the PI material, which leads to its poor thermal conductivity (approximately 0.1 W/(m·K)). It is difficult to achieve rapid heat dissipation for pure PI. Accumulated heat will deepen the internal temperature gradient of the material, thereby causing electric field distortion, which is unfavourable to material surface insulation [7,8,9,10]. On the other hand, environmental factors, such as intense electric fields and the generation of reactive species in plasma by corona discharge, will also lead to insulation ageing or even the failure of PI [11,12]. Therefore, in recent years, research on improving the thermal conductivity of PI to achieve rapid heat dissipation while also improving its insulation stability in various working environments has received considerable attention.

This is an effective way to obtain high-performance composite materials by doping specific reinforcement materials into a polymer matrix. To improve the heat dissipation ability of PI materials, some inorganic filling materials are commonly used, including alumina (Al_2_O_3_), aluminium nitride (AlN), boron nitride (BN), silicon nitride (Si_3_N_4_), silica (SiO_2_), silicon carbide (SiC), along with other materials, such as nanosheets and nanotubes [13,14,15,16,17,18,19]. Among them, hexagonal boron nitride (h-BN), known as “white graphene”, has high thermal conductivity (up to 380 W/m·K) and a low dielectric constant (approximately 4), which makes it a good filler with comprehensive properties [20]. Sombel Diaham et al. studied the thermal conductivity of two sizes of boron nitride nanocomposite films (40 nm and 120 nm); moreover, they studied the thermal conductivity of boron nitride nanocomposite films with different contents [21]. The results showed that the particle size of BN had a great influence on the thermal conductivity of PI. The thermal conductivity of PI/h-Bn (120 nm) increased to 0.56 W/(m·K) when the filling volume percentage was 29.2 vol%. Steve et al. innovatively dispersed both nanoscale and microscale boron nitrite particles in a polyimide matrix in different proportions and mass fractions and obtained a PI/BN composite material with differently sized dopants [22]. The thermal conductivity was up to 1.2 W/(m·K) in the 30 wt% doping system with a micro-nanometre weight ratio of 7:3.

Compared with BN nanoparticles, boron nitride nanosheets (BNNSs) are a two-dimensional, thermally conductive filler. The interfacial thermal resistance of PI/BNNS can be significantly reduced when BNNSs are arranged in a certain direction inside the PI matrix. A thermally conductive network chain parallel to the direction of heat flow can be formed in the composite material, which significantly reduces phonon scattering in the material. Therefore, BNNSs are also widely used in thermally conductive composite materials [20]. Wang et al. introduced BNNSs into PI. The results showed that 7 wt% filler could increase the in-plane thermal conductivity to 2.95 W/(m·K) and the out-of-plane thermal conductivity to 0.44 W/(m·K) [23]. Using hydrophobic BNNSs as the skeleton and silver nanowires (AgNWs) as the “thermally conductive bridge”, Dong et al. attempted to construct a separate 3D nanometre-scale thermally conductive network with a specific orientation in a PI matrix to improve the phonon transmission capacity and reduce the interfacial thermal resistance [19]. A PI/BNNS composite with a doping ratio of 20 wt% was prepared, and its thermal conductivity reached 4.75 W/(m·K), which was 324% higher than that of pure PI. This thermal conductivity was significantly better than that of the randomly doped PI/BNNS system.

According to the above studies, it is apparent that the intrinsic thermal conductivity of PI can be effectively improved using a BN filling, especially two-dimensional oriented BNNSs. Most current studies prepare PI/BNNS with different design configurations and then test the physical and chemical properties of the obtained materials. However, few studies have focused on the microscopic mechanism of how BNNSs improves the thermal conductivity of composites and inhibits the structural damage of the PI matrix at high temperatures. Moreover, the insulation material is bound to withstand an intense electric field during application, which may trigger corona discharge and generate reactive species. The intense electric field and activity of the plasma particles can also damage the PI insulation material, leading to its failure. Therefore, to further explore the insulation failure mechanism of a PI/BNNS material, appropriate means need to be selected to simulate the effects of high temperature, an intense electric field, and the presence of reactive species in plasma on PI/BNNS composites at the microscale level.

With the rapid development of computer technology, molecular simulations, especially molecular dynamics (MD) simulation methods, are widely used for the research and design of polymer materials. As a new generation of molecular dynamics force fields, the reactive force field (ReaxFF) combines the advantages of quantum mechanics and traditional dynamics, which not only ensures the accuracy of calculation but also makes it possible to quickly calculate a large system and reflect the dynamic path of microscopic reactions. It is the most commonly used simulation method for the research of material properties. Yu et al. studied the effect of differently sized nanoalumina particles on the mechanical properties of epoxy resin using molecular dynamics simulation and found that the small nanoalumina particle filling resulted in an epoxy resin composite with higher mechanical properties [24]. Mayank et al. explored the effects of different overhanging polar groups on the dielectric loss of a polyethylene copolymer through molecular dynamics simulation and proposed the time–temperature superposition (TTS) method, providing a more rapid and efficient method for calculating dielectric loss [25]. Therefore, we can use molecular simulation to effectively study and analyze the microscopic reaction mechanism of PI/BNNS materials at high temperatures, in intense electric fields, and with reactive species in the plasma.

To further clarify the inhibitory effect of a BNNSs filling on PI insulation damage in different environments, a PI/BNNS composite model was established based on ReaxFF. The decomposition of the PI matrix and PI/BNNS at high temperatures, in intense electric fields, and with reactive species were simulated. The microscopic mechanism of the inhibitory effect of BNNSs on PI insulation failure was investigated at the atomic level. This paper provides a theoretical basis for studying the PI insulation ageing mechanism and developing new high thermal conductivity insulation materials.

## 2. Principle of ReaxFF

ReaxFF is a reaction field associated with bond order, which can reflect polarization and charge transfer effects in complex systems. ReaxFF can describe strong interactions, such as covalent bonds and ionic bonds, as well as nonbonded interactions, including van der Waals forces and Coulomb forces; therefore, it can accurately simulate the process of chemical bond cleavage and formation. With the characteristics of a small amount of calculation and high precision that is close to that of the quantum mechanics method, ReaxFF acts as a bridge between quantum mechanics and molecular dynamics [26,27,28]. ReaxFF determines whether two atoms are connected at the current moment based on the relationship between the bond order (BO) and the bond distance (BD), bond level and energy and then represents the cleavage and formation process of interatomic chemical bonds [29,30,31,32].

ReaxFF defines the interatomic bond level as a function of atomic spacing. The spacing of two atoms (marked i and j, respectively) at a certain time is r_ij_, and the uncorrected bond level is shown in the following formula.
(1)$$\begin{array}{cc} BO_{ij}^{\prime } & =BO_{ij}^{\sigma }+BO_{ij}^{\pi }+BO_{ij}^{\pi \pi }\\ & =\exp\left[p_{\mathrm{bo}1}g{\left(\frac{r_{\mathrm{ij}}}{r_{0}^{\sigma }}\right)}^{p_{\mathrm{bo}2}}\right]+\exp\left[p_{\mathrm{bo}3}g{\left(\frac{r_{\mathrm{ij}}}{r_{0}^{\pi }}\right)}^{p_{\mathrm{bo}4}}\right]+\exp\left[p_{\mathrm{bo}5}g{\left(\frac{r_{\mathrm{ij}}}{r_{0}^{\pi \pi }}\right)}^{p_{\mathrm{bo}6}}\right] \end{array}$$
where BO_ij_^σ^, BO_ij_^π^, and BO_ij_^ππ^ represent a single bond, double bond, and triple bond, respectively; r_0_^σ^, r_0_^π^, and r_0_^ππ^ are atomic parameters; p_bo_ is the bond parameter. Relationship between the atomic distance and bond formation and cleavage is shown in Table 1.

On this basis, to correct the bond level, a function Δ′, which represents the overcoordination number of the central atom, is introduced, as shown in Formula (2):(2)$$\Delta_{i}^{\prime }{=-\mathrm{Val}}_{i}+\overset{\mathrm{neighbours}\left(i\right)}{\underset{j=1}{\sum }}{\mathrm{BO}}_{\mathrm{ij}}^{\prime }$$
where Val_i_ is the total bond order of atom i, and the neighbours are collections of atoms with which atom i forms a valence. For the corrected bond order of BO_ij_, BO_ij_ = 0.3BO_ij_ is taken as the criterion of whether the chemical bond breaks. Notably, BO_ij_ ≥ 0.3BO_ij_ represents that a chemical bond is formed; otherwise, the chemical bond is considered to be broken.

Similar to the traditional force field, ReaxFF divides the total energy of molecules into the sum of different parts, and the energy of each part is a function of the bond level. The total energy of the system is expressed as follows:(3)$$E_{\mathrm{system}}{=E}_{\mathrm{bond}}{+E}_{\mathrm{over}}{+E}_{\mathrm{under}}{+E}_{\mathrm{val}}{+E}_{\mathrm{pen}}{+E}_{\mathrm{tors}}{+E}_{\mathrm{conj}}{+E}_{\mathrm{vdWaals}}{+E}_{\mathrm{Coulomb}}$$
where E_bond_ is the bond energy, E_over_ and E_under_ are the energy correction terms of overcoordination, E_val_ is the valence angle energy term, E_pen_ is the penalty energy term, E_tors_ is the torque energy term, E_conj_ is the conjugate action term of the molecule, E_vdWaals_ is the term of nonbonded van der Waals forces, and E_Coulomb_ is the term of the nonbonded Coulomb forces.

## 3. Force Field and Modelling

Currently, the most commonly used solid insulating PI material is homophenyl polyimide [33]. This PI material is formed by the condensation polymerization and dehydration reaction of diaminodiphenyl ether and homophenyl dihydride. The molecular formula is (C_22_H_10_N_2_O_5_)_n_, where n is the degree of polymerization (DP). The structure is shown in Figure 1.

Above all, a reasonable force field should be selected based on the reaction system to specify parametric relationships between different atoms. According to the elements contained in PI, BNNSs and reactive species (·H_3_O, ·NO, ·O_3_, ·OH, etc.) that may be involved in the reaction, the HCONSB.ff force field was selected in this study for subsequent simulation according to relevant literature [34,35,36].

Based on the molecular structure of the PI monomer, a reasonably pure matrix PI model was first constructed (PI neat). Relevant studies showed that the DP of the PI long chain did not affect the law of PI cleavage. In consideration of the calculation period, PI molecules with a DP of four were selected for decomposition simulation in this paper [37]. To establish a material model that was closest to the actual situation, exploratory tests were carried out to determine the size of the periodic amorphous cell system and the number of PI long chains. Finally, a periodic amorphous cell system model of 35 × 35 × 35 Å was established, and five PI long chains were distributed in the model to construct the PI neat system.

To verify the reliability of the model, crucial parameters of PI neat were calculated. The relative dielectric constant and thermal conductivity of PI neat were approximately 3.3 and 0.124 W/(m·K), respectively, while these two parameters of the experimentally measured PI were approximately 2.9–3.8 and 0.11 W/(m·K), respectively [38,39,40]. The calculated results were consistent with the experimental data, which showed that the PI neat model built above could accurately describe the actual PI matrix and could be used to further study the mechanism of PI decomposition. The PI/BNNS decomposition models were constructed by filling a 10% mass fraction of BNNSs in the above model. Periodic boundary conditions were used in the simulation, and the simulation was repeated five times. Therefore, the dispersion problem was not considered when filling with BNNSs.

Before the decomposition simulation, the models needed to be pretreated. First, the energy minimization function was used to optimize the system. Then, the NVT ensemble (constant atomic number, constant volume, and constant temperature) was adopted to conduct the kinetic optimization of the minimized energy model at 300 K for 50 ps. In the second half of the optimization, the total energy of the system tended to a stable value, indicating that the molecular conformation of the system was more reasonable after the above treatment. The optimized decomposition system is shown in Figure 2.

## 4. Decomposition Simulation and Analysis of the PI Neat and PI/BNNS Systems

### 4.1. Pyrolysis Caused by High Temperature

The pyrolysis of a material at high temperatures is a long-term process. A low temperature in the simulation will cause a major expenditure of time and greatly increase the computational cost. According to the temperature-accelerated reaction kinetics, appropriately increasing the temperature can advance the initial time of the reaction while not affecting the molecular dynamics behaviour of the whole reaction process. Therefore, a temperature higher than the actual condition is normally chosen in simulation to accelerate the reaction process [41]. In this paper, pyrolysis simulations were carried out based on the two systems constructed above for 100 ps at 2000 K, 2200 K, 2400 K, 2600 K, 2800 K, and 3000 K. The NVT ensemble was used, and the time step was 0.1 fs. The calculations were repeated five times for each group.

Relevant studies showed that the characteristic small-molecule products of PI pyrolysis include carbon dioxide (CO_2_) and cyanide (CN) [42]. Therefore, in this paper, the pyrolysis degree of the PI molecule was determined by observing the molecular chain fracture of PI and the amount of produced CO_2_ and CN in each reaction system. When the temperature was 2000 K, the molecular chain of PI in the two systems twisted, but few PI chains broke. Only a small number of small-molecule products were generated, as shown in Figure 3. In the PI/BNNS system, the BNNSs were slightly distorted; however, the original two-dimensional sheet structure was basically maintained, with part of the PI winding around the BNNSs.

As the simulated temperature was increased, the pyrolysis of the PI neat and PI/BNNS systems became increasingly obvious. For example, at 2200 K, the amount of generated CO_2_ was higher than at 2000 K, as shown in Figure 4. In the PI/BNNS system, the BNNSs deformed further as the temperature was increased. Of significance, the PI long chain attached to the edge of the BNNSs did not break easily, indicating that the presence of BNNSs did have a certain inhibitory effect on PI cleavage.

The pyrolysis results at 3000 K are shown in Figure 5. It is not hard to note that the cleavage of PI molecules in both systems was very serious; nearly all the PI long chains were broken. Moreover, large amounts of CO_2_ and CN were generated. At this temperature, the BNNSs in the PI/BNNS system were severely deformed and fractured. The inhibition of the fractured BNNSs on PI pyrolysis weakened; however, aggregated PI still existed.

By observing the reaction process, it can be seen that the C-N bond in the imide ring was the first to break at a high temperature, as shown in Figure 6. The C=O bond and C=C bond gradually broke to produce CO_2_, which was consistent with the PI pyrolysis path in the existing literature, proving that the simulation results in this paper were reasonable [42,43].

To quantitatively analyze the inhibitory effect of the BNNS filling on the decomposition of the PI matrix at high temperatures, appropriate quantitative evaluation parameters were introduced. Since the initial atomic number and the molecular number of the two systems were different, it was not precise in judging the decomposition degree solely by the molecular number of each system. Therefore, the fragmentation index was defined in this paper to reflect the decomposition degree of different systems. The fragmentation index was defined as N_0_, the total number of molecular fragments in each model was N_m_, the initial atomic number was N, and the fragmentation index was calculated by the following formula:(4)$$N_{0}=\frac{N_{m}}{N}$$

The fragmentation index of the PI neat system after high-temperature pyrolysis at 2000–3000 K is shown in Figure 7a. As the temperature was increased, the fragmentation index continued to increase, and the generation of pyrolysis products gradually increased. Thus, the increase in temperature resulted in more complete pyrolysis of PI.

The fragmentation index of the PI neat system and PI/BNNS system at 2800 K and 3000 K are shown in Figure 7b. There was no significant difference in the fragmentation index between the two systems in the initial 50 ps. At 100 ps, the fragmentation index of PI neat at 2800 K was 0.077, while that of PI/BNNS was approximately 0.068. The number of molecules decreased by approximately 12%. At 3000 K, the fragmentation indices of PI and PI/BNNS were 0.1 and 0.08, respectively. The number of molecules in the doped system decreased by approximately 20%.

To further analyze the inhibition effect of BNNS doping on PI pyrolysis, the number of crucial small-molecule products (CO_2_ and CN) of each system at 2000 K, 2200 K, 2400 K, 2600 K, 2800 K, and 3000 K were counted at 100 ps, as shown in Figure 8. ∑C is the sum of CO_2_ and CN.

The number of small-molecule products gradually increased as the temperature was increased, indicating the gradual intensification of the decomposition degree of the PI material. From 2000 K to 3000 K, the total number of small-molecule products from the PI/BNNS system was smaller than that from the PI neat system, indicating that the BNNSs significantly inhibited the pyrolysis of PI at high temperatures. This could be interpreted by the strong interaction between the BNNSs and PI molecules, which restricted the movement of PI molecules around the BNNSs. Additionally, the stable network structure of the filler not only improved its heat resistance but also partly prevented the movement of PI molecules, effectively inhibiting the breaking of bonds between atoms. Therefore, the presence of the BNNSs increased the energy required for the pyrolysis of the surrounding PI and inhibited the thermal ageing of the PI material.

### 4.2. Decomposition Caused by an Intense Electric Field

In a strong electric field environment, insulating materials will be pulled by electric field stress. Moreover, a strong electric field will also cause surface discharge of materials, resulting in deformation and even fracture of the molecular structure. Relevant experimental data show that the initial discharge voltage of the PI film was approximately 5 kV/mm, and the breakdown field strength was approximately 200~400 kV/mm [44]. In this paper, different electric fields were applied to the two optimized models for simulation. To fully study the influence of different field intensities on material decomposition, the simulated field intensity interval was selected near the breakdown strength of PI. PI and PI/BNNS were simulated in three electric fields; namely, 100, 200, and 400 kV/mm. The simulation duration was set to 50 ps, and the step size was set to 0.1 ps.

The variation in the molecular numbers in the PI neat and PI/BNNS systems over time in an electric field with the same order of magnitude are shown in Figure 9a,b, respectively. It is obvious that the variation trend and final stability number of molecules in the two systems are nearly the same at the three electric field strengths. In the PI/BNNS system, the number of molecules in a 100 kV/mm electric field is lower than that in the 200 kV/mm and 400 kV/mm electric fields. Notably, the difference is not significant, indicating that there is little difference in the PI decomposition degree at the same electric field level. Therefore, to facilitate the observation of simulation results, this paper increased the gap between the different simulated electric field intensities. Electric field intensities of 5 kV/mm, 50 kV/mm, and 200 kV/mm were selected for the simulation calculations to summarize and analyze the influence of the BNNS filling on the decomposition of PI due to an intense electric field that is within the field intensity range of the initial discharge field intensity to the breakdown field intensity.

First, the PI neat system was taken as an example to qualitatively study the influence of the electric field intensity on the material decomposition degree. Based on the above, the electric field intensity was determined to be 5 kV/mm, 50 kV/mm, and 200 kV/mm, and the results are shown in Figure 10a. The number of molecules in the system continued to increase, and the decomposition degree of PI deepened with increasing electric field intensity. This also indicated that the PI material was gradually destroyed when the electric field was increased from the initial discharge strength to the breakdown strength, which meant that the material insulation properties gradually degraded. To study the inhibition effect of BNNS doping on the decomposition of materials in different electric fields, the decomposition degrees of the material at 5 kV/mm, 50 kV/mm, and 200 kV/mm were further compared.

To accurately describe the PI and PI/BNNS systems with different initial molecular numbers, fragmentation indices of the different electric field decomposition systems were calculated. The fragmentation index of each system at 50 ps at electric field intensities of 5 kV/mm, 50 kV/mm, and 200 kV/mm is shown in Figure 10b. As seen from the figure, the fragmentation index of the PI/BNNS system was slightly lower than that of the PI system at the initial discharge field intensity, indicating that the BNNS filling could inhibit material decomposition in an electric field. In fact, due to the difference between the density of the simulation system and the actual material density, the inhibitory effect was not particularly obvious. However, related studies showed that in actual composite materials, the BNNS filling could significantly improve the thermal conductivity of a pure polymer [44]. The heat generated by the corona discharge could be distributed effectively, and the increase in the relative dielectric constant with temperature was restrained to a certain extent, thereby reducing the charge accumulation on the insulation material surface. Thus, the corona resistance of the BNNS composite was improved. All these results proved the validity of the conclusion in this paper to a certain extent.

However, with the increase in electric field intensity, the fragmentation index of the PI neat system at 50 ps was lower than that of the PI/BNNS system at 50 kV/mm and 200 kV/mm, indicating that the BNNS filling reduced the upper limit of resistance to an intense electric field (that is, the breakdown field intensity of the material decreased). This reflected the inevitable problem regarding the doping of composite materials; that is, the inorganic material filling will inevitably lead to the appearance of internal defects in composite materials. When the mass fraction of inorganic filler was increased, more organic–inorganic interface traps were formed inside the composite material. This resulted in more ion accumulation, which led to partial electric field distortion and a decrease in the breakdown field strength. Therefore, when the mass fraction of BNNSs was 10%, the breakdown strength of PI/BNNS was lower than that of pure PI materials.

To screen out a PI/BNNS material with better comprehensive performance, the decomposition of PI/BNNS materials with different doping ratios at breakdown electric field intensities was further simulated and analyzed. The influence of the BNNS mass fraction on the inhibition of composite decomposition in an electric field was studied. The modelling method of the PI/BNNS models with different mass fractions was similar to that of the PI/BNNS−10% system; therefore, the details will not be explained here. The variation trend of the molecular number in the PI/BNNS systems with different doping ratios at an electric field intensity of 200 kV/mm is shown in Figure 11. As seen from the figure, the number of molecules in the system increased when the doping ratio was increased, which could be explained by the following two aspects. On the one hand, the difference in the initial BNNS size in different systems affected the number of small molecules produced by BNNS decomposition in an electric field. On the other hand, according to our previous studies, a lower doping ratio was more conducive to improving the breakdown strength of polymer-based composites [45]. To quantitatively measure the decomposition degree of each system, systems with different doping ratios were subjected to 200 kV/mm for 50 ps, and their fragmentation index was calculated in this paper (Figure 12).

When the BNNS filling mass fraction was below 5%, the fragmentation index of the doped system was lower than that of the undoped system. At this stage, the BNNSs acted as a skeleton. The PI molecules attached to the BNNSs could resist the pull of the electric field and were less likely to deform. In addition, the high-frequency molecular movement caused by the electric field resulted in a large amount of heat inside the material, and the improvement in the thermal conductivity of the composite material could transfer this heat in time to prevent local overheating and breakdown. At this stage, the doped BNNSs were beneficial for improving the breakdown strength of PI composites.

As the mass fraction of BNNSs was increased, a large amount of BNNSs would lead to an excessive accumulation of ions inside the material, which would cause electric field distortion and affect the insulation performance of the material. Furthermore, plentiful inorganic–organic phase interfaces were formed between the PI matrix and BNNSs due to the high doping ratio. Some interfaces might overlap to form conductive paths, which would easily lead to the breakdown of the PI composite films. Therefore, it could be seen from the figure that when the mass fraction exceeded 5%, the fragmentation index at 50 ps increased with an increasing BNNS mass fraction. In this case, the breakdown field intensity of the doped system was lower than that of the pure PI system, which put forward higher requirements for the precise control of the BNNS doping ratio in the preparation of composites.

### 4.3. Decomposition Caused by Reactive Species in Plasma

Corona discharge and partial discharge are inevitable during the operation of high-voltage power equipment. The reactive species in plasma produced by air ionization during corona discharge impact and erode the surface of insulating materials, which is one of the main reasons for insulation material deterioration. Therefore, the effect of the reactive species in plasma should also be considered when studying the decomposition of insulating materials.

The accumulation and distribution of space charge in corona discharge are different, so corona discharge has two different forming mechanisms—positive polarity corona and negative polarity corona. Wayne et al. found that the erosive effects of ·H_3_O and ·NO were most obvious among reactive species produced by the positive polarity corona in air [46]. Chen and Pavlik et al. studied negative polarity corona plasma in dry and wet air, respectively, and the results showed that ·O_3_ had the highest content and that ·OH particles had the strongest activity [47,48]. Therefore, when studying the influence of AC corona plasma on PI composites, the influences of the above four particles are particularly important.

PI and PI/BNNS models were also established in the MD simulation with reactive species. ·H_3_O, ·NO, ·O_3_, and ·OH particles were separately placed into the two systems to study the effects of the four particles on the PI composites. The inhibitory effect of the BNNS filling on the erosion of materials by the reactive species in plasma was analyzed. Ten reactive particles were added to each reaction system. The energy of each system was minimized before the reaction simulation started. During the reaction, the MD simulation was performed for 100 ps under the NVE ensemble (constant atomic number, constant volume, and constant energy), and the reaction step was set to 0.1 ps.

First, the influence of various reactive particles on the PI neat system and PI/BNNS system was analyzed, and the total molecular numbers of each system within 100 ps were counted (Figure 13a,b). When the four kinds of reactive particles acted on the PI system, the total numbers of ·H_3_O and O_3_ systems were greater than those of the other systems in the initial 40 ps, and the material decomposition phenomenon was obvious. However, in the ·OH and ·NO systems, the number of molecules increased slowly or even decreased to a certain extent, indicating that the reactive particles adsorbed on the PI matrix; however, they did not decompose the material into small molecules. From 40 ps to 70 ps, the molecular numbers of all systems increased rapidly. Most of the PI molecular chains broke, and a large number of small molecules were formed. At 100 ps, the number of small molecules in the ·NO system was significantly higher than that in other systems, while the number of molecules in the ·H_3_O system was slightly lower than that in other systems. The numbers of small molecules in the ·O_3_ and ·OH systems were basically the same.

When the four kinds of reactive particles acted on the PI/BNNS system, the total molecular number of the ·O_3_ system was always slightly higher than that of the other systems. Both the growth trend of the total molecular numbers and the final molecular numbers were basically the same as those of other systems. In the PI/BNNS system, the total molecular number of the ·OH and ·NO systems decreased slightly in the first 30 ps, which was similar to the situation in the PI system. This indicated that in the PI/BNNS system, ·OH and ·NO were also absorbed on the matrix. However, it was obvious that the adsorption effect of the PI/BNNS system was not as significant as that of the PI system.

To compare the decomposition degree of the PI neat system and PI/BNNS system under the influence of reactive species, the fragmentation index of the different systems at 100 ps was calculated, as shown in Figure 14.

The fragmentation index of the PI/BNNS system was higher than that of the PI system when affected by ·H_3_O and ·O_3_. Additionally, the BNNS filling had no inhibitory effect on the ageing of the materials. The fragmentation index of the PI/BNNS system was lower than that of the PI system when affected by ·OH and ·NO. Moreover, the BNNS filling was helpful in alleviating the decomposition of materials exposed to plasma.

To further study the influence of the discharge corona types on material decomposition, the coronas were classified according to the generated reactive species. The fragmentation index of each system under the effects of the different corona types was counted, as shown in Figure 15. It can be seen from the figure that the erosive effect of the positive corona on the material was more obvious than that of the negative corona. The relevant experimental results showed that the reactive species produced by the positive corona discharge caused more damage to the polymer than those produced by the negative corona discharge [49]. This conclusion could also explain why the fragmentation index of the positive corona was higher than that of the negative corona. On the other hand, the fragmentation index of the PI/BNNS system was always slightly lower than that of the PI system for both corona discharges. Overall, with the AC corona, the BNNS filling could inhibit the decomposition caused by the reactive species in the plasma.

From the molecular number perspective, the interaction process of reactive species and materials was roughly analyzed above. However, the specific microscopic process and reaction pathway of reactive particles on materials were not clarified. To further explore the microscopic mechanism of the effect of various reactive particles on the material, the specific process of H_3_O, ·NO, ·O_3_, and ·OH particles acting on the PI material are explored at the atomic level and described below.

#### 4.3.1. Functional Mechanism of ·H_3_O

The functional mechanism of ·H_3_O could be attributed to hydrogen capture and adsorption. First, due to the unstable, reactive structure of ·H_3_O, the hydrogen atoms in ·H_3_O would be continuously captured by PI molecules at the initial stage of the reaction between ·H_3_O and PI, as shown in Figure 16a,b. Compared with the carbon atoms, the oxygen and nitrogen atoms in the PI molecular chains were more likely to capture hydrogen from ·H_3_O, thus forming -OH and an unstable -NH structure. Moreover, the reactive particles were still free, so the number of molecules in the system did not decrease.

·H_3_O was dehydrogenated to form water molecules, which would be further adsorbed on the nitrogen atoms of the PI chain, as shown in Figure 16c,d. After a series of transition state reactions, the reactive groups mentioned above eventually became reactive oxygen species connected to the PI carbon atoms, causing the fracture of the C-N bond. Nitrogen atoms combined with free hydrogen atoms to form small NH_2_ molecules. The adsorption of oxygen atoms further promoted the decomposition of the PI benzene ring structure and the generation of small-molecule products, such as CO_2_. This was why the number of molecules increased steadily in the ·H_3_O system. However, the H_2_O generated by the dehydrogenation of ·H_3_O was relatively stable, and no violent reaction occurred during the subsequent adsorption process, which led to a low final fragmentation index when the PI system was exposed to ·H_3_O.

#### 4.3.2. Functional Mechanism of ·NO

When reacting with PI molecules, a large amount of ·NO would be adsorbed on the oxygen atoms in the PI molecular chains at the initial stage of the reaction, as shown in Figure 17a. Therefore, there was a decrease before the increase in the number of molecules in the system. After a period of the transition state reaction, ·NO began to be adsorbed on the carbon atoms of the PI molecular chains, as shown in Figure 17b. When absorbed on carbon atoms, ·NO provided enough energy to cause the chain to break, breaking the fragile C-N bond and generating small-molecule products, as shown in Figure 18. Thus, the total number of molecules in the system started to increase. The self-reaction of ·NO was not as strong as that of ·H_3_O, so more ·NO would participate in the reaction with PI, which also made the fragmentation index of the ·NO system significantly higher than that of other systems. The self-reaction of ·NO is shown below.
(5)$${\mathrm{NO}}^{+}{+\mathrm{NO}}^{+}\to N_{2}{+O}_{2}$$

N_2_ generated by the reaction above had a stable structure and had almost no participation in the subsequent reaction. The remaining O_2_ could continue to oxidize the PI structure; however, the reaction was not as violent as the ·NO treatment. The fragmentation index of the system exposed to ·NO was significantly higher than that of other systems due to the weak self-reaction of the reactive particles and the dual effects of ·NO and O_2_.

#### 4.3.3. Functional Mechanism of ·O_3_

·O_3_ was mainly adsorbed on the carbon atoms of the PI molecular chain, and its functional mechanism is shown in Figure 19. ·O_3_ was first adsorbed on the carbon atom in the imide ring, leading to the fracture of the unstable C-N bond in the imide ring, as shown in Figure 19a. There was an intermediate -N-O-C- structure, as shown in Figure 19b. Moreover, an O-O bond in ·O_3_ broke, and a stable O_2_ structure dissociated from it. Then, the intermediate structure N-O bond broke, as shown in Figure 19c. At this time, the imide ring was completely fractured, and then the C-C bond broke; thus, the small molecule CO_2_ was formed (Figure 19d). The active chemical properties of ·O_3_ led to violent self-reactions. Furthermore, O_2_ molecules would be generated in the adsorption process of ·O_3_; therefore, the number of molecules did not decrease at the beginning of the reaction.

The self-reaction of ·O_3_ generated a large amount of O_2_. Although O_2_ could also react with the PI molecular chain, it was far less active than ·O_3_. Therefore, the strong self-reaction of ·O_3_ resulted in a limited effect on the decomposition of the PI molecular chain, and the increase in the molecular number of the system was limited to a certain extent.

#### 4.3.4. Functional Mechanism of ·OH

The functional mechanism of ·OH was similar to that of ·O_3_, as shown in Figure 20. ·OH was adsorbed on the carbon atoms in the imide ring, resulting in the breaking of the C-N bond on the imide ring; notably, this was also the first fracture position in the PI molecular chain, as shown in Figure 20a. When ·OH was adsorbed on the imide ring, the hydrogen atom in ·OH was captured by the oxygen atom in the imide ring, as shown in Figure 20b. The carbon atom adsorbing ·OH was linked to two oxygen atoms to obtain the energy required for breaking bonds. Finally, the C-C bond was broken, and the carbon atom was dissociated from the imide ring to form a small molecular product CO_2_, as shown in Figure 20c. The initial reaction of the ·OH system was similar to that of ·O_3_; however, O_2_ would not be further generated for the secondary reaction. Therefore, the number of molecules in the system would decrease at the initial stage of the reaction, and the number of molecules in the ·O_3_ system would be slightly more than that in the ·OH system as the molecular number continued to increase.

Additionally, some hydrogen capture occurred in the ·OH system, as shown in Figure 21. ·OH grabbed the hydrogen atom from the benzene ring of PI to form the small molecule H_2_O. The carbon atom without a hydrogen atom was easier to combine with the oxygen atom, leading to the structure of the benzene ring becoming unstable. The hydrogen capture effect of ·OH promoted the reaction to a certain extent. Not only adsorption but also hydrogen capture occurred in the ·OH system. Therefore, the inflection point of the molecular number curve was earlier than that of the ·NO system, where there was only an adsorption process. However, the number of molecules decreased in the early reaction stage of the ·OH system.

It is worth noting that the BNNSs hardly reacted with any reactive species in the plasma. The stable structure of the BNNSs made it difficult for nitrogen atoms to capture hydrogen atoms from the reactive particles. It was also difficult for oxygen atoms in the reactive species to be adsorbed or react on the BNNSs. Therefore, the BNNSs could inhibit the erosion of composites by reactive species in the plasma. However, the BNNSs might fracture with the twisting of molecules in the system, and the ability of the BNNSs to inhibit decomposition caused by reactive species in plasma would be greatly weakened. In fact, the BNNSs could also inhibit the self-reaction of reactive species, according to the simulation results. Regarding the ·O_3_ system with the most self-reaction, the BNNSs could inhibit ·the erosion of the PI material by ·O3, while they also reduced a large amount of the self-consumption reaction of ·O_3_; thus, the activity of the ·O_3_ particles was maintained. Therefore, the molecular numbers of the PI/BNNS groups in the ·O3 system were significantly higher than those in the other systems.

In fact, when corona discharge occurs on the surface of insulating materials, the high-energy electrons generated by corona discharge continue to bombard the surface of insulating materials. Under normal circumstances, when the main chain remains intact, and the molecular structure is relatively stable, the damaging effect of electron bombardment on the material is limited. However, the reactive species in plasma (·H_3_O, ·NO, ·O_3_, and ·OH) produced under different corona discharges can activate the C-O and C-N bonds in the main chain and even the C-C bonds in the benzene ring, causing the main chain to break and become an unstable structure that is more vulnerable to electron attack. Meanwhile, due to the existence of hydrogen abstraction, the surface-inert methine (-CH) and imino group (-NH) are destroyed, which not only makes the molecular structure of the material become unstable but also reduces the hydrophobicity of the material. This triggers more intense chain reactions under corona discharge. An avalanche will eventually be caused, resulting in the overall deterioration of the comprehensive properties of the insulating material.

## 5. Conclusions

In this paper, PI neat and PI/BNNS composite insulation models were built. Molecular dynamics simulations were carried out under conditions that insulation materials might face in practical work: high temperatures, intense electric fields, and various reactive species in the plasma.

(1)According to the simulation results of pyrolysis at 2000 K~3000 K, the decomposition degree of the PI material and the number of generated small molecules all gradually increased with increasing temperature. At different temperatures, the number of small-molecule products in the PI/BNNS system was always smaller than that in the PI neat system. On the one hand, the BNNSs inhibited the fracture of the molecular chain by limiting the movement of the PI long chain. Moreover, the doping of BNNSs significantly improved the resistance of PI composites to pyrolysis.(2)To study the decomposition processes from initial corona discharge to breakdown, electric field strengths of 5 kV/mm, 50 kV/mm, and 200 kV/mm were selected for simulation. With increasing electric field intensity, the number of molecules in the system increased, and PI molecular chain cleavage became increasingly thorough. A 10 wt% BNNS filling had a certain inhibitory effect on material decomposition at the initial discharge field intensity but reduced the breakdown strength of the material to a certain extent. A low concentration of the BNNS filling was more conducive to improving the breakdown strength of the material and preventing electric field distortion caused by excessive ion aggregation.(3)To study the material decomposition caused by reactive species in plasma, ·H_3_O and ·NO generated by the positive corona and ·O_3_ and ·OH generated by the negative corona were selected for molecular dynamics simulation. In pure PI, the number of small molecules in the ·NO system was significantly higher than that in other systems. In the PI/BNNS groups, the number of molecules in the ·O_3_ system was always slightly higher than that in the other systems. The BNNS filling could inhibit the decomposition process of materials caused by ·OH and ·NO. According to the analysis of different corona conditions, the erosive effect of the positive corona on PI composites was more obvious, and the BNNS filling could partially inhibit the decomposition of the PI composite caused by plasma.(4)The functional mechanisms of various reactive species on PI/BNNS were analyzed at the atomic level, which mainly included the process of hydrogen capture and adsorption. The effect of reactive particles on materials mainly depended on the degree of particle activity and the degree of self-reaction. The generated H_2_O molecules had weak activity. The self-reaction of ·O_3_ was violent, resulting in a large amount of self-consumption. There was no excessive self-consumption reaction in the ·NO system, and the O_2_ generated in the ·NO system could further oxidize PI molecules. Therefore, among the studied reactive particles, ·NO had the most damaging effect on the composite materials. There was almost no reaction between the reactive particles and BNNSs. However, the BNNSs reduced the self-consumption of reactive particles, thus keeping the reactive particles active.

## Figures and Tables

**Figure 1 polymers-14-01169-f001:**
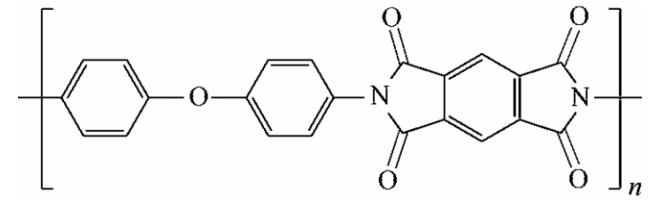
Molecular structure of polyimide.

**Figure 2 polymers-14-01169-f002:**
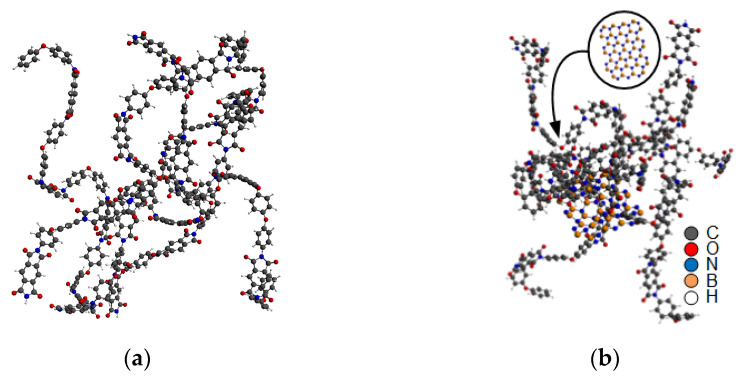
Optimized decomposition model of PI and PI/BNNS. (**a**) PI neat; (**b**) PI/BNNS.

**Figure 3 polymers-14-01169-f003:**
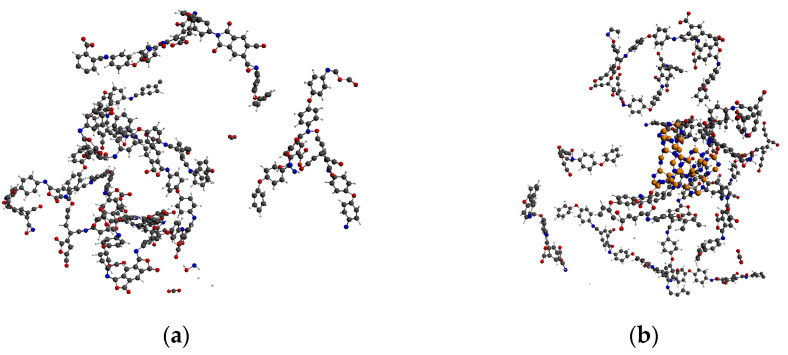
Pyrolysis results at 2000 K. (**a**) PI neat; (**b**) PI/BNNS.

**Figure 4 polymers-14-01169-f004:**
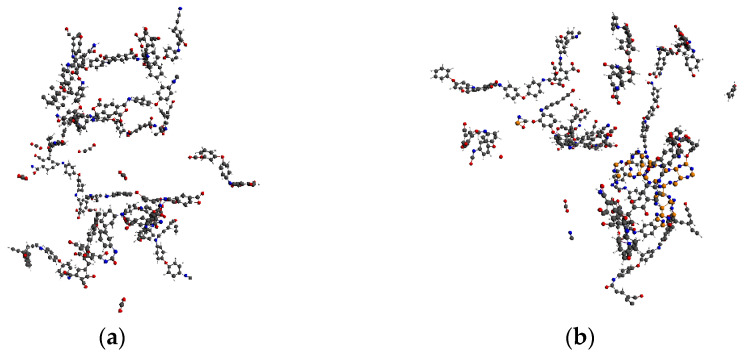
Pyrolysis results at 2200 K. (**a**) PI neat; (**b**) PI/BNNS.

**Figure 5 polymers-14-01169-f005:**
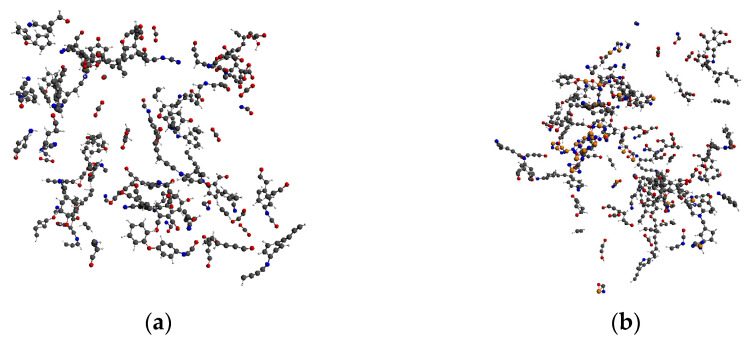
Pyrolysis results at 3000 K. (**a**) PI neat; (**b**) PI/BNNS.

**Figure 6 polymers-14-01169-f006:**
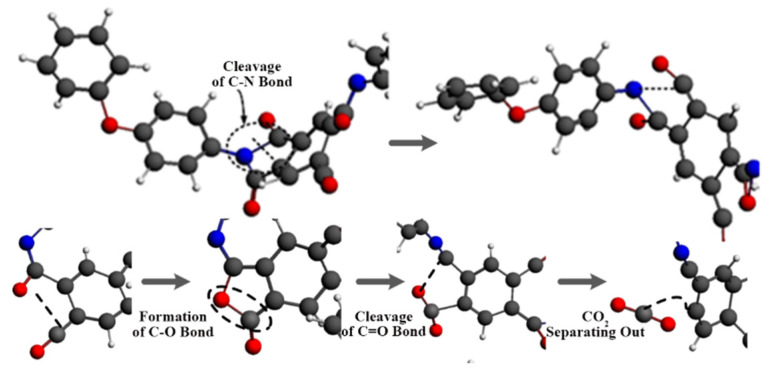
Microscopic process of PI pyrolysis.

**Figure 7 polymers-14-01169-f007:**
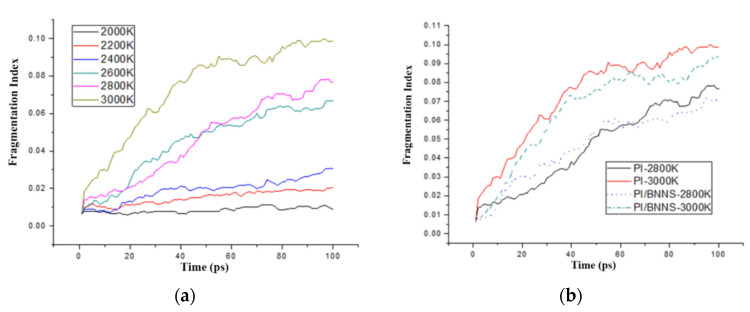
(**a**) PI neat fragmentation index after pyrolysis at high temperatures; (**b**) PI neat and PI/BNNS fragmentation index after pyrolysis at high temperatures.

**Figure 8 polymers-14-01169-f008:**
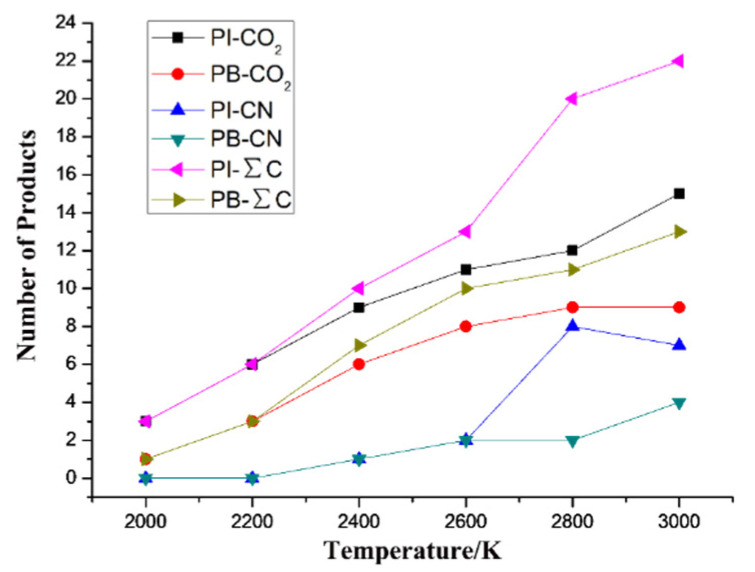
Product formation of the PI neat and PI/BNNS systems at different temperatures.

**Figure 9 polymers-14-01169-f009:**
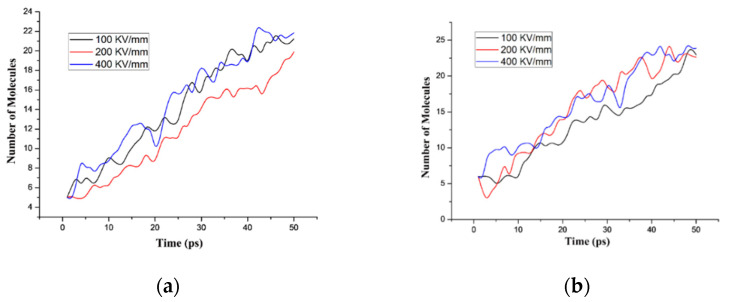
Molecular number of the different systems at three electric fields with the same order of magnitude. (**a**) PI neat; (**b**) PI/BNNS.

**Figure 10 polymers-14-01169-f010:**
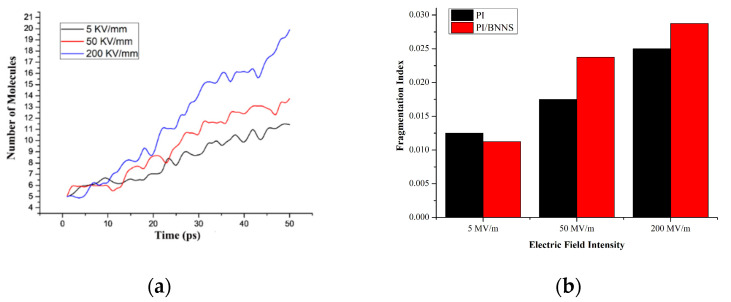
The decomposition degree of different systems under different electric field intensities. (**a**) Molecular number of the PI neat system at different electric field intensities; (**b**) Fragmentation index at 50 ps of the PI neat and PI/BNNS systems at different electric field intensities.

**Figure 11 polymers-14-01169-f011:**
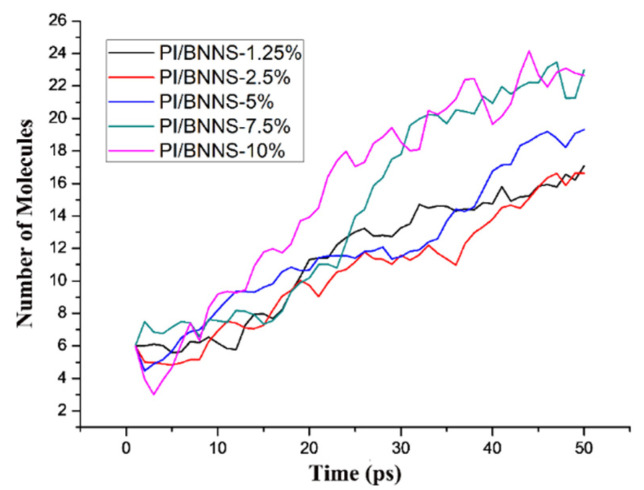
Variation in the molecular number over time for PI/BNNS systems with different doping ratios at a 200 kV/mm field intensity.

**Figure 12 polymers-14-01169-f012:**
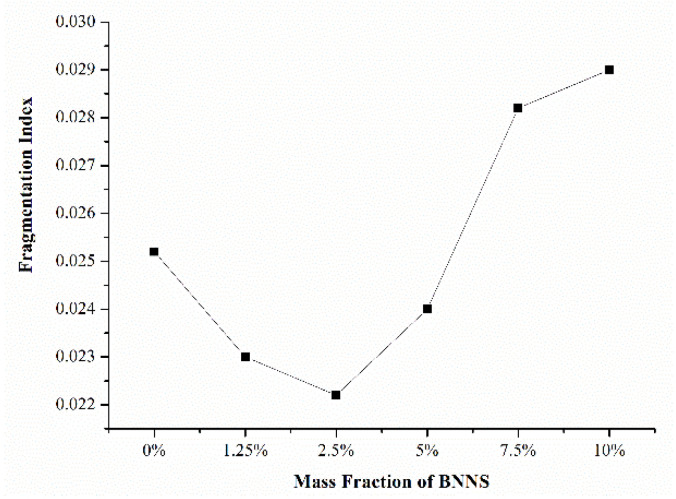
Fragmentation index of PI/BNNS systems with different doping ratios at a 200 kV/mm field intensity.

**Figure 13 polymers-14-01169-f013:**
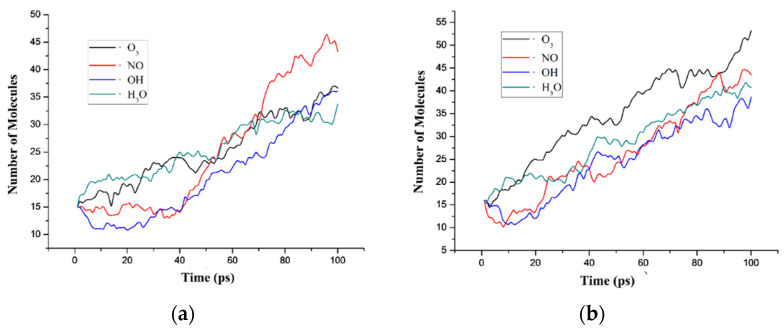
Number of molecules in different systems treated by reactive species. (**a**) PI neat; (**b**) PI/BNNS.

**Figure 14 polymers-14-01169-f014:**
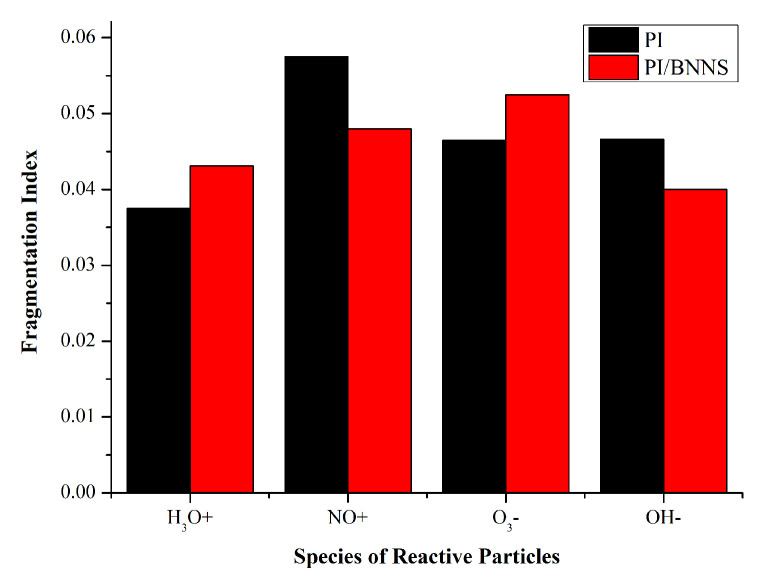
Fragmentation index of the different systems at 100 ps.

**Figure 15 polymers-14-01169-f015:**
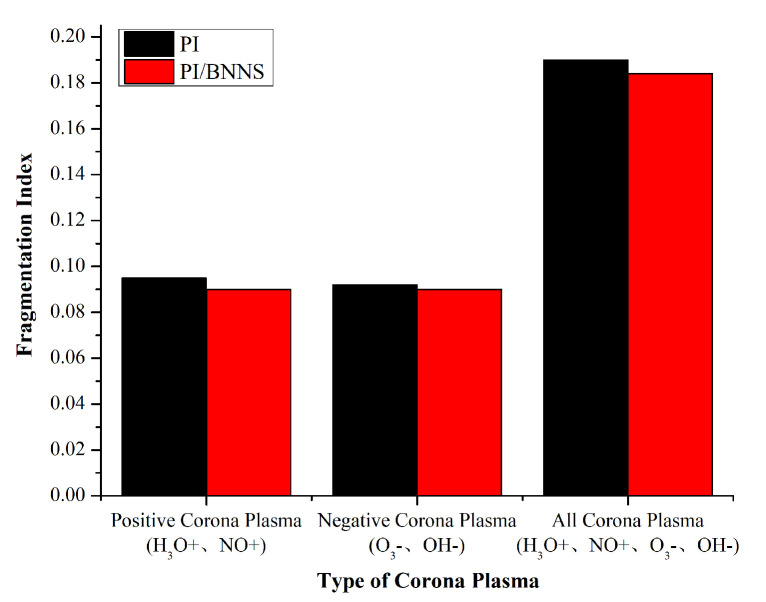
System fragmentation index under the effect of the different corona types at 100 ps.

**Figure 16 polymers-14-01169-f016:**
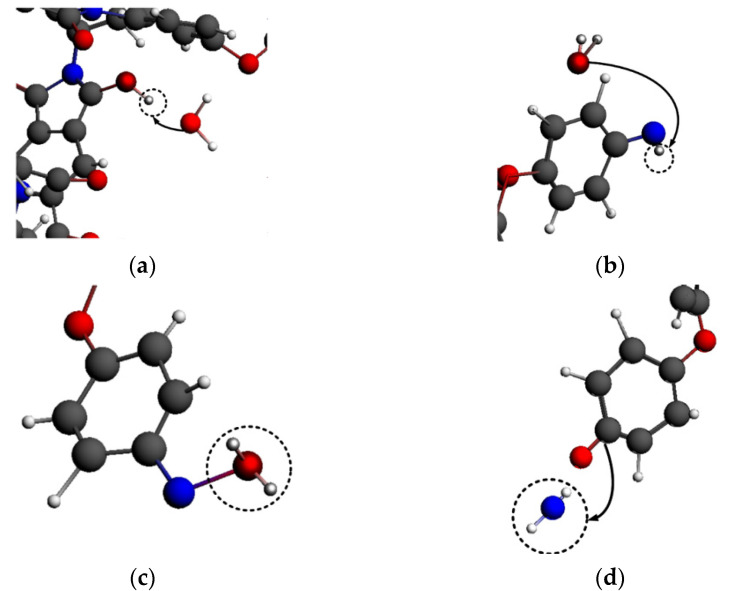
Functional mechanism of ·H_3_O (**a**) hydrogen abstraction process of oxygen; (**b**) hydrogen abstraction process of nitrogen; (**c**) adsorption process of ·H_3_O; (**d**) adsorption process of ·H_3_O.

**Figure 17 polymers-14-01169-f017:**
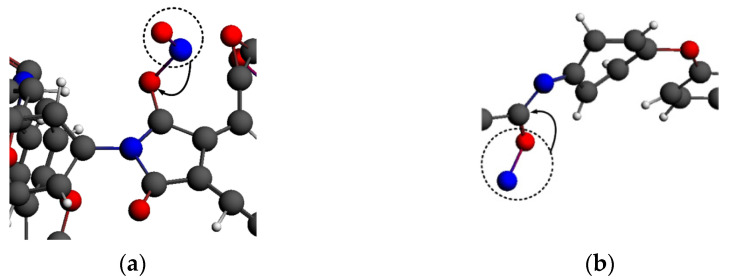
Adsorption process of ·NO. (**a**) ·NO adsorbed on the PI oxygen atoms; (**b**) ·NO adsorbed on the PI carbon atoms.

**Figure 18 polymers-14-01169-f018:**
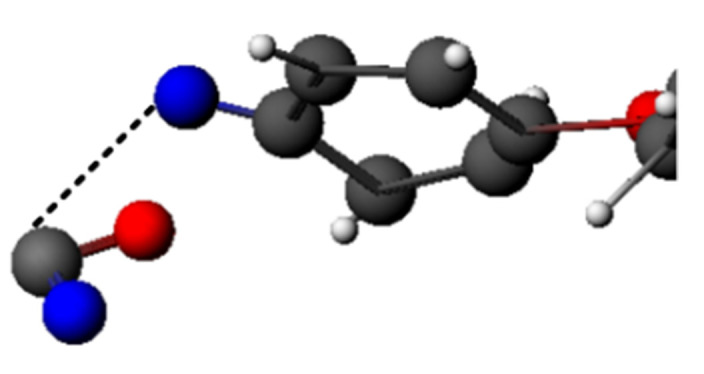
Breaking process of the C-N bond in the PI chain due to ·NO treatment.

**Figure 19 polymers-14-01169-f019:**
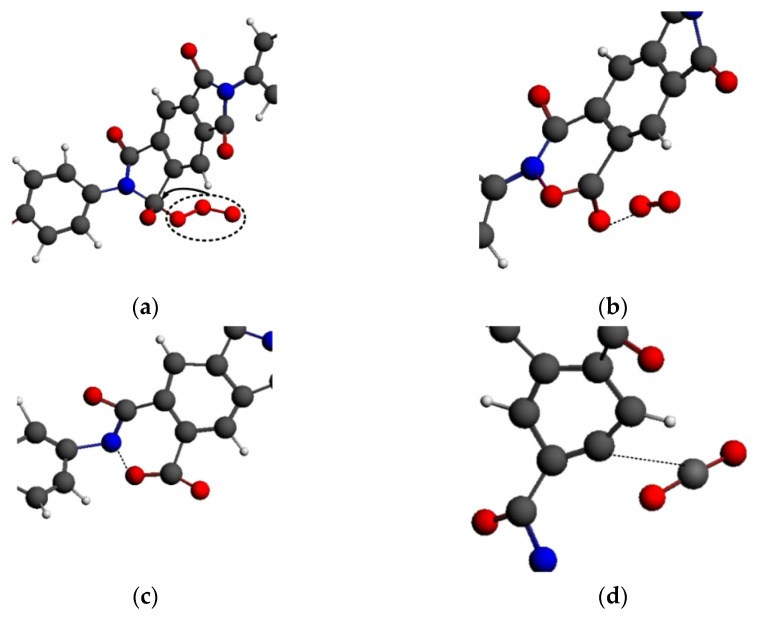
Functional mechanism of ·O_3_. (**a**) ·Oxygen adsorbed on the PI carbon atoms; (**b**) The exfoliation of oxygen molecules; (**c**) The breaking of nitrogen-oxygen bond; (**d**) The exfoliation of CO_2_ molecules from main chain.

**Figure 20 polymers-14-01169-f020:**
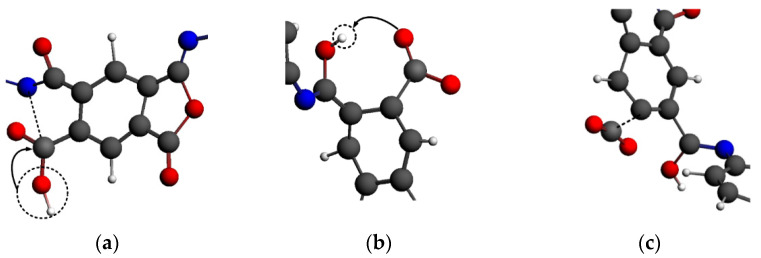
Adsorption process and mechanism of ·OH. (**a**) The addition reaction of hydroxyl; (**b**) The transfer of hydrogen; (**c**) The exfoliation of CO_2_ molecules from main chain.

**Figure 21 polymers-14-01169-f021:**
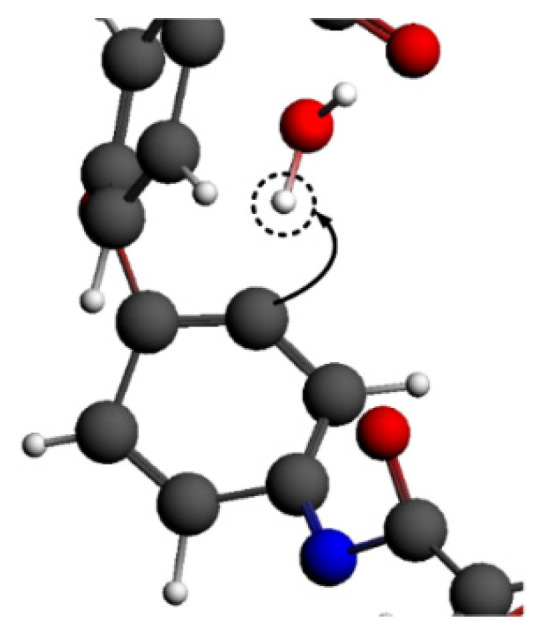
Hydrogen abstraction mechanism of ·OH.

**Table 1 polymers-14-01169-t001:** Relationship between the atomic distance and bond formation and cleavage.

Type	$BO_{\mathrm{ij}}^{\sigma }\;(\mathrm{Single}\;\mathrm{Bond})$	$BO_{\mathrm{ij}}^{\pi }\;(\mathrm{Double}\;\mathrm{Bond})$
Bond formation	$r_{\mathrm{ij}}$ < 1.5 Å	$r_{\mathrm{ij}}$ < 1.2 Å
Bond cleavage	$r_{\mathrm{ij}}$ > 2.5 Å	$r_{\mathrm{ij}}$ > 1.75 Å

## Data Availability

The data that support the findings of this study are available upon reasonable request from the authors.

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
