# Peer review of "Microscopic Pyrolytic and Electric Decomposition Mechanism of Insulating Polyimide/Boron Nitride Nanosheet Composites based on ReaxFF"

_polymers, 2022, doi:10.3390/polym14061169_

Round 1
Reviewer 1 Report
The molecular dynamics simulations were carried out at high temperatures, in intense electric fields and with various reactive species in plasma based on the reactive force field (ReaxFF). The main purpose of the simulation is to study the microscopic mechanism of composite material decomposition in an actual working environment and the inhibitory effect of BNNS doping on the decomposition process. The results showed that the decomposition was mainly caused by hydrogen capture and adsorption, which broke the benzene ring and C-N bond on the PI chains and led to serious damage to the PI structure. The BNNS filling inhibits the decomposition of the PI matrix at high temperatures and in intense electric fields. Moreover, the BNNS filling also inhibited the material decomposition caused by ·OH and ·NO. However, the microscopic dynamic reaction paths of material pyrolysis in various environments were revealed at the atomic level, and it was concluded that BNNS doping could effectively inhibit the decomposition of PI in various environments. The work is interesting and written well with supporting data. So, the paper can be published after minor revision.
The figure format and arrangement in the manuscript are not well. A lot of figure number were mentioned. It is better to reduce smaller number of figures. It means that the current figure can be merged as a Figure 1 (a, b, c, d) instead of Figure 1(a, b) and Figure 2(c, d). For example, Figure 17 and Figure 18 cab be merged as Figure 17 (a, b). Similarly, Figure 7 and Figure 8 can be merged as Figure 7(a, b). Same thing can be applicable for the remaining Figure 9~22.
Author Response
Dear Referee,
Thank you for your comments concerning our manuscript entitled “Microscopic Pyrolytic and Electric Decomposition Mechanism of Insulating Polyimide/Boron Nitride Nanosheet Composites based on ReaxFF” (ID: polymers-1627603). The comments were valuable and very helpful for revising and improving our paper as well as an important guide for the significance of our research. We have studied the comments carefully and have made corrections that we hope will be met with approval. The main corrections in the paper and responses to the referee’s comments are as follows:
Comment :
The figure format and arrangement in the manuscript are not well. A lot of figure number were mentioned. It is better to reduce smaller number of figures. It means that the current figure can be merged as a Figure 1 (a, b, c, d) instead of Figure 1(a, b) and Figure 2(c, d). For example, Figure 17 and Figure 18 cab be merged as Figure 17 (a, b). Similarly, Figure 7 and Figure 8 can be merged as Figure 7(a, b). Same thing can be applicable for the remaining Figure 9~22.
Response:
Thank you very much for your valuable comments and suggestions.
We are very sorry for our less rigorous figure arrangement. Based on your comments, we have rearranged figures in our manuscript.
In the revised paper, we have merged Figure 7 and Figure 8 as Figure 7(a, b), Figure 10 and Figure 11 as Figure 9(a, b), Figure 12 and Figure 13 as Figure 10(a, b), Figure 16 and Figure 17 as Figure 13(a, b), Figure 20(a, b) and Figure 21(a, b) as Figure 16(a, b, c, d). The numbers of other figures have also been revised accordingly. We hope our reply can satisfy you.
Thank you again for your valuable comments.

Reviewer 2 Report
The work «Microscopic Pyrolytic and Electric Decomposition Mechanism of Insulating Polyimide/Boron Nitride Nanosheet Composites based on ReaxFF» is devoted very interesting topic. Regulation of the functional properties of polymeric materials by varying the type and concentration of filler in composites is a rapidly developing area of scientific research. The choice of the object of this study is well justified in the introduction. Simulation of material decomposition under various conditions will make it possible to predict the actual behavior of the polymer and composite. Discussion is supported by informative and vivid figures. But I have comment:
The information presented on the mechanisms of the decomposition reactions of polyamide and the composite based on it does not fully reflect the processes of chain reactions, which are the basis of the corona discharge. Please complete the description.
Author Response
Dear Referee,
Thank you for your comments concerning our manuscript entitled “Microscopic Pyrolytic and Electric Decomposition Mechanism of Insulating Polyimide/Boron Nitride Nanosheet Composites based on ReaxFF” (ID: polymers-1627603). The comments were valuable and very helpful for revising and improving our paper as well as an important guide for the significance of our research. We have studied the comments carefully and have made corrections that we hope will be met with approval. The main corrections in the paper and responses to the referee’s comments are as follows:
Comment:
The information presented on the mechanisms of the decomposition reactions of polyamide and the composite based on it does not fully reflect the processes of chain reactions, which are the basis of the corona discharge. Please complete the description.
Response:
Thank you very much for your valuable comments and suggestions.
The chain reaction in the corona discharge process is caused by high-energy electron bombardment to cause the fracture of the fragile bonds in the material molecules, which leads to the collapse of the molecular stability and finally leads to the avalanche, resulting in the overall deterioration of the material properties. In fact, the reactive species in plasma produced under different corona discharges (·H3O, ·NO, ·O3 and ·OH ) can activate the C-O and C-N bonds in the main chain and even the C-C bonds in the benzene ring, causing the main chain to break and become an unstable structure that is more vulnerable to electron attack. Meanwhile, due to the existence of hydrogen abstraction, the surface-inert methine (-CH) and imino group (-NH) are destroyed, which not only makes the molecular structure of the material become unstable, but also reduces the hydrophobicity of the material. This triggers more intense chain reactions under corona discharge.
The part “Decomposition caused by reactive species in plasma” in our manuscript was focused on analysis of the damage mechanism caused by the reactive species in plasma produced by corona discharge. The influence of the existence of different particles on the subsequent chain reactions under the same conditions was analyzed. Therefore, only the head reaction induced by plasma active particles was analyzed, but the chain reaction under different discharge conditions under the action of high-energy electrons was not analyzed. Therefore, only the head reactions triggered by reactive species in plasma were analyzed. The subsequent chain reactions under different discharge caused by high-energy electrons were not analyzed separately. However, your comments are very valuable. Chain reaction is an important path of material destruction under corona discharge. The relation between erosion effect of reactive species in plasma and chain reaction in corona discharge was lacking in the original manuscript.
In the revised paper, we have added relevant explanations and descriptions in lines 587-600. We hope our reply can satisfy you.
Thank you again for your valuable comments.
